# Examination of Non-Specific Low Back Pain, Pain Perceptions and Disability Between Brazilian Jiu Jitsu, Muay Thai and Boxing Athletes

**DOI:** 10.3390/healthcare13050447

**Published:** 2025-02-20

**Authors:** Anna Christakou, Elena Karvouni, Ioannis S. Benetos, Dimitrios S. Evangelopoulos, Spyridon G. Pneumaticos

**Affiliations:** 1Laboratory of Biomechanics, Department of Physiotherapy, University of Peloponnese, 23100 Sparta, Greece; 2Department of Physiotherapy, University of West Attica, 12243 Athens, Greece; 33rd Department of Orthopaedic Surgery, KAT Hospital, National and Kapodistrian University of Athens (NKUA), 16541 Athens, Greece; karvouniel@hotmail.com (E.K.); ioannisbenetos@yahoo.gr (I.S.B.); ds.evangelopoulos@gmail.com (D.S.E.); spirospneumaticos@gmail.com (S.G.P.)

**Keywords:** martial arts, low back pain, pain perceptions, disability

## Abstract

**Background:** Non-specific low back pain is the leading cause of years lived with disability worldwide. The present study investigates non-specific low back pain, pain perceptions and disability due to pain among Brazilian Jiu Jitsu, Muay Thai and Boxing athletes. **Methods:** The study included 90 amateur athletes (aged 18–45 years; M = 28.97, SD = 5.88). The athletes completed the valid and reliable Pain Beliefs Perceptions Inventory (PBPI), the Quebec Pain Disability Scale (QPDS) and the Short-Form McGill Pain Questionnaire (SF-MPQ) which includes the Visual Analogue Scale (10 cm VAS 0–10 rating system) and the Present Pain Intensity index (PPI). Results: The results revealed that the majority of athletes rated their pain as low (SF-MPQ: M = 12.34, SD = 8.91; VAS: M = 1.65, SD = 1.82; PPI: M = 2.10, SD = 1.08) with low disability due to pain (QPDS: M = 18.98, SD = 22.71). Also, the majority of athletes disagreed that their pain was mysterious or persistent with high duration (PBPI: M = 1.43, SD = 2.23). Between the three martial arts, Brazilian Jiu Jitsu athletes showed statistically significantly (a) higher emotional and sensational pain intensity (x^2^(2) = 15.73; *p* < 0.001; x^2^(2) = 19.34; *p* < 0.001), (b) higher disability due to pain (x^2^(2)= 25.30; *p* < 0.001) and (c) more mysterious, more persistent pain with more duration (x^2^(2)= 9.32; *p* < 0.05) than Muay Thai and Boxing athletes. Also, a few correlations were found between age and pain perception only in Brazilian Jiu Jitsu and Boxing martial arts athletes. **Conclusions:** Further research is required to elucidate the biomechanical and psychological factors contributing to these differences between martial arts athletes.

## 1. Introduction

Low back pain (LBP) is the leading cause of years lived with disability worldwide. In clinical trials, LBP is often poorly categorized into ‘chronic’ versus ‘acute’ pain [1]. Chronic low back pain (CLBP), defined as lumbar pain persisting for 12 weeks or more, is a widespread and persistent health issue in the general population, recognized as the leading cause of disability in developed countries [2]. Hoy et al. [3] reported that CLBP affects a significant proportion of individuals, i.e., about 13% of U.S. adults, with 85–90% of people expected to experience it at some point in their lives, and 2–5% suffering from it annually [4]. It is known to have a more complex etiology and is influenced by a wider range of factors, including social, psychological and cultural elements, that can have a more profound impact on chronic than acute LBP [5]. Acute LBP differs from chronic low back pain in terms of duration and underlying factors. It is characterized by the sudden or gradual onset of pain in the lower back, lasting from a few days up to four weeks. Patients often describe the pain as stabbing, tearing or cutting; it sometimes radiates into the lower extremities, producing “sciatic-like” symptoms. Importantly, acute and chronic LBP lead to a significant reduction in quality of life, limited participation in daily activities and decreased work capacity [6].

Athletes are often subjected to repetitive physical stress, making them more vulnerable to LBP [7]. For athletes, particularly those engaged in high-demand sports, LBP can severely affect performance, leading to long-term disability and reducing training effectiveness [8]. The most common causes of lumbar pain include inadequate biomechanical patterns, muscle imbalance, improper warm-ups, technical errors and inadequate training [9]. Also, Loeser et al.’s [10] study highlights the interconnectedness of biological, psychological and social factors in the experience of pain. These biopsychosocial exposures are a complex interplay and gaining insight in understanding the pathways has profound implications for chronic LBP. The exposures that have been recognized as having a crucial role in CLBP are job control, demand, strain, support, stress and satisfaction [11]. In particular, the high demands in the sport domain, the athletes’ stress level and satisfaction status may influence his/her training and competitive season, and thus may cause physiological alterations such as pain. As a result, it is important to examine the sensory and emotional factor of pain between athletes with LBP. Pain assessment is crucial for successful rehabilitation and several instruments, such as the Pain Beliefs Perceptions Inventory [12], the Short-Form McGill Pain Questionnaire [13] and the Quebec Pain Disability Scale [14], are commonly used to evaluate pain perceptions, beliefs and disability due to pain in populations with musculoskeletal pain.

Studies have examined the prevalence, impact and treatment of LBP in athletic populations [15,16]. Cross-sectional studies reported the association between swimming and LBP, but this relationship was not confirmed recently by Wareham et al.’s scoping review [17]. Ansari and Sharma [18] reported a significant correlation between CLBP, sleep quality and chronotype in university’s athletes with CLBP. Also, Taekwondo athletes with non-specific LBP participated in a three-direction movement control focus complex pain program, and they improved the stability control ability of their lumbar spine [19].

However due to the lack of research on LBP in martial artists, the aim of this study was to investigate pain, pain perceptions and disability due to pain among athletes from Muay Thai, Brazilian Jiu Jitsu, and Boxing martial arts. These three martial arts have different disciplines that employ distinct movements—Brazilian Jiu Jitsu with its focus on grappling, Muay Thai with its striking techniques, and Boxing with its combination of footwork and punching—each potentially leading to different patterns of low back pain and disability [20,21,22]. Studies on LBP in these specific athletic populations remain scarce, making it difficult to understand how these athletes cope with pain, how their training might exacerbate or mitigate it and if it affects their functional ability in everyday activities [23,24,25].

No study has investigated so far the pain perceptions and disability due to pain between the Brazilian Jiu Jitsu, Muay Thai and Boxing athletes. The aim of this study was to investigate pain, pain perceptions and disability due to pain among athletes and the relationship of pain perception with age in the three aforementioned martial arts. Identifying any pain differences will not only help in developing sport-specific preventive strategies, but also contribute to creating more effective treatment and rehabilitation programs tailored for these specific martial artists. Therefore, we aim to address a gap in the existing literature and offer new insights into the impact of LBP on Brazilian Jiu Jitsu, Muay Thai and Boxing athletes, ultimately improving their everyday functional ability and their sport performance.

## 2. Materials and Methods

### 2.1. Design

This study was a cross-sectional research design. It has been registered and approved by the Ethics Committee of the Medical School (No 10/2023, 11 January 2023). The study was in agreement with the Declaration of Helsinki ethics principles. The outcomes of the study included the completion of 3 valid questionnaires by the athletes.

### 2.2. Participants

The participants were informed about (a) the aim of the study, (b) the voluntary nature of participation, and (c) the confidentiality of their responses. Athletes registered to the study (convenience sample) were asked to sign an informed consent document. They had the right to terminate their participation from the study at any time and they were informed that any publication of the results would be anonymous. The athletes were directly contacted by the 2nd author.

The sample consisted of 90 athletes (46 men, 44 women) (30 boxers, 30 athletes from Muay Thai, 30 from Brazilian Jiu Jitsu), aged from 18 to 45 years old (M = 28.97, SD = 5.88). The inclusion criteria were (a) amateur athletes; (b) aged between 18 and 40 years, regardless of sex; (c) active in the sport at least three times weekly during the past two years; (d) non-specific low back pain during the last 3 months [26] with a diagnosis by a physician.

Exclusion criteria for the sample included the following: (a) history of persistent back and low back pain before participating in sports, (b) scoliosis, (c) congenital spinal disorders, (d) history of lumbar spine surgery, (e) spondylolisthesis, spinal infection, vertebral fracture, systemic disease or treatment for the lumbar spine through the use of an MRI scan by a physician, (f) use of analgesic, anti-inflammatory, or muscle relaxant drugs within the previous week and (g) presence of any rheumatological or cardiovascular disease, chronic neurological or psychiatric disorders, drug addiction, anemia or diabetes.

### 2.3. Measurements and Procedure

Three valid instruments were used in the study:


(a)
*The Short-Form McGill Pain Questionnaire (SF-MPQ)*



The SF-MPQ is a widely recognized instrument that evaluates the influence of pain on an individual’s psychological and physical well-being. Patients are requested to evaluate each pain descriptor in accordance with the level of distress they are currently experiencing. In particular, the SF-MPQ consists of 15 descriptors (11 sensory; 4 affective) which are rated on an intensity scale as 0 = none, 1 = mild, 2 = moderate or 3 = severe. The SF-MPQ also includes the Present Pain Intensity (PPI) index (with a 0–5 rating system) and a 10 cm visual analogue scale (VAS with 0–10 rating system). The SF-MPQ has been translated into Greek by Georgoudis et al. [13]. The SF-MPQ takes approximately 5 min to complete and score.


(b)
*The Pain Beliefs Perceptions Inventory (PBPI)*



The PBPI is a 16-item self-administered Likert-type scale addressing 4 dimensions of pain beliefs: seeing pain as mysterious (Mystery subscale), holding oneself responsible for pain (Self-Blame subscale) and regarding one’s condition as lingering in the future (Permanence subscale) and/or continuous over time (Constancy subscale). Each item had a score ranged from −2 to +2, without a 0 value. Items 3, 9, 12, and 15 were reverse-scored. A total score was obtained by dividing the total sum by the number of items. Higher scores indicate greater endorsement of the beliefs and perceptions. The initial version of the PBPI has been shown to exhibit satisfactory acceptability, excellent reliability and good reproducibility; furthermore, its content validity and construct validity has been established in patients with pain. The inventory has been validated in the Greek population [12].


(c)
*The Quebec Back Pain Disability Scale (QBPDS)*



The QBPD scale is a 20-item self-administered instrument designed to assess the level of functional disability in individuals with back pain. The twenty items are classified into six areas: rest/bed, sit/stand, ambulation, handling of large/heavy objects, movement and bending/stooping. Each item relates to a specific activity which can be scored from 0 (no difficulty at all) to 5 (unable to do). The maximum total score is 100. The scale has been validated in the Greek population [14].

Additionally, the athletes completed a questionnaire with demographical data and the above three instruments before the daily training. The duration to complete the three questionnaires was approximately 10 min.

### 2.4. Statistical Analysis

The homogeneity between groups was assessed using an independent samples t-test for numerical variables and the chi-square (χ^2^) test for categorical variables with a significance level of α = 0.05. Descriptive statistics tests were performed using the means and standard deviations of the sample. A normality test was performed using the Kolmogorov–Smirnov test. The Kolmogorov–Smirnov test revealed that all variables deviated significantly from a normal distribution; thus, Kruskal–Wallis H comparisons were performed between groups. Also, we examined the relationship between pain perception and age in the three martial arts using Pearson r analysis. All analyses were conducted using the SPSS statistical package (version 29.0.0) (IBM Corporation, Somers, NY, USA).

## 3. Results

Table 1 and Table 2 provide detailed demographic information for the sample of 90 athletes across Brazilian Jiu Jitsu, Muay Thai and Boxing martial arts. The age for Brazilian Jiu Jitsu athletes ranged from 20 to 37 years old (M = 28.33, SD = 5.31), for Muay Thai from 19 to 45 years old (M = 28.73, SD = 7.30) and for Boxing athletes from 21 to 40 years old (M = 21.40, SD = 4, 85). The gender distribution of the sample was nearly balanced, i.e., 51.1% of the athletes were male and 48.9% were female athletes. Although pain perception could be affected by gender, no differences were observed between groups. The educational level revealed that the athletes were fairly educated, with 54.4% having a university level education and 23.3% with other qualifications. This may affect their perception of pain and how it should be treated. Other lifestyle factors include marital status and the presence of children, and these variables did not differ between the three groups. The study revealed that the athletes trained 3–4 times a week (41.1%) and for 2 h daily (71.1%). Regarding training interruptions due to pain, 64.4% (*n* = 58) of athletes did not miss training, while 15.6% (*n* = 14) missed 2 months, 11.1% (*n* = 10) missed 1 month, and smaller proportions missed varying amounts of training time (3–5 months).

Table 3 displays the mean, standard deviation, minimum and maximum scores of the PPBI, the QBPDS and the SF-MPQ for the 90 athletes from the three martial arts. For the PPBI, the overall average score was −1.43 (SD = 2.23), which means that the sample had lower endorsement of their pain beliefs and perceptions; for the QBPDS, the total mean score was 18.98 (SD = 22.71), which means having low disability due to pain. The total mean score of the SF-MPQ was 12.34 (SD = 8.91), which indicates low pain intensity among athletes.

Table 4 illustrates the comparison of pain, pain beliefs/perceptions and disability among Muay Thai, Brazilian Jiu Jitsu and Boxing athletes. There are statistically significant differences between the three groups of athletes in almost all the measurement variables. In particular, Brazilian Jiu Jitsu athletes showed statistically significantly (a) the highest emotional and sensory pain intensity (x^2^(2) = 15.73; *p* < 0.001; x^2^(2) = 19.34; *p* < 0.001, respectively), (b) the highest disability due to pain (x^2^(2)= 25.33; *p* < 0.001) and (c) the most perception of mysterious, persistent and high-duration pain (x^2^(2)= 9.32; *p* < 0.05) compared to Muay Thai and Boxing athletes. In contrast, Boxing athletes showed (a) the lowest emotional and sensory pain intensity (M = 1.93, SD = 1.78; M =7.97, SD = 5.86, respectively), (b) the lowest disability due to pain (M = 7.57, SD = 8.42) and (c) the lowest perception of mysterious, persistent with no duration pain (M = −2.22, SD = 2.15) compared to Brazilian Jiu Jitsu and Muay Thai athletes.

We also found a correlation between age and pain perception in the Visual Analogue Scale in Brazilian Jiu Jitsu athletes (r = −0.42, *p* < 0.05) and between age and the sensory dimension of the Short-Form McGill Pain Questionnaire (total) in Boxing athletes (r = −0.41, *p* < 0.05).

## 4. Discussion

The present study was designed to evaluate the differences on the LBP, pain beliefs/perceptions and disability due to pain and the relationship between pain perception and age in Brazilian Jiu Jitsu, Muay Thai and Boxing athletes. The results of the study showed that there was disagreement amongst the athletes whether their pain was mysterious or persistent with high duration as assessed by the valid PBPI. They did not attribute their distress to themselves and they disagreed that a perpetual experience of suffering would be a lifelong burden for them. Comparing the three martial arts, Brazilian Jiu Jitsu athletes significantly experienced their pain as mysterious and persistent with high duration. In contrast, the Boxing athletes had the lowest endorsement of the pain beliefs among the three martial arts. It is unclear why Brazilian Jiu Jitsu athletes had greater endorsement of pain beliefs in comparison to the other two martial arts, although their pain objectively was rated as low. The unique grappling and ground techniques characteristic of Brazilian Jiu Jitsu, which place significant stress on the lumbar spine, are probably responsible for these pain beliefs. No other study has investigated the pain perceptions between different martial arts athletes with LBP. Thus, future studies should confirm the present results and examine more the relationship of pain beliefs between different martial arts including Judo and Karate.

Athletes in the present study have low disability due to pain as measured by the valid QBPDS. Similarly, another study found that low back pain had little effect on Karate practitioners’ functional status [27]. Self-reported functional disability is one of the most important outcome measures in back pain research. The QBPDS appears to be the most useful measure of functional outcomes for people with LBP. Moreover, if an athlete has limitations in daily activities, we believe they should have the same limitations in sports activities. However, Brazilian Jiu Jitsu athletes again experienced the most disability due to pain in comparison with the other two martial arts. In contrast, the Boxing athletes had the lowest disability among the three martial arts. This disability level by Brazilian Jiu Jitsu athletes may be linked to the martial art’s characteristics, such as the principle of defeating the adversary by using their own force, which contributes to significant stress on their lumbar spine. Violent movements like kick and punches are replaced by projections, grasping, joint locks and chokes which they may believe that affect their functional status. Brazilian Jiu Jitsu athletes probably believe that their pain may provoke disability in their everyday life activities. Due to the lack of studies, more research should be conducted to confirm the present results.

Martial arts are characterized by intense training. The length of training and the need to retain mental focus, combined with the all-too-frequent shortening of the time needed for tissue regeneration, can place a significant overload on the lumbar spine. However, all the athletes in the present study for the three martial arts presented low pain as assessed by SF-MPQ, VAS and PPI index. Similarly, 45% of Karate practitioners experience quite little pain (3–4 VAS) [19]. Ridan et al. [9] found similar results, i.e., an average of 3.7 VAS in mixed martial arts fighters. On the contrary, other studies reported that judokas had moderate low back pain [28,29]. Yabe et al. [30] reported that LBP was highest in Judo, followed by Kendo and Karate. Older age was associated with LBP in Judo, Kendo and Karate while lower extremity pain was associated with LBP only in Judo and Kendo. In studies with wrestlers, Granhedi and Moreli [31] reported a prevalence of CLBP in 59% of athletes; Lundinet al. [32] found that 56% of their sample had CLBP. Another study among Kyokushin Karate practitioners has shown that practitioners with 2–5 years of training experience represent 45% of all respondents with LBP [27]. Reis et al. [33] reported that chronic LBP was present in 80.6% of Jiu Jitsu athletes. In their study, pain was present in 88.9% of professional and 35.6% of recreational athletes. The present study did not investigate differences between demographical characteristics like competition level, sex and age. Also, in the present study, statistically significant differences were seen across the three martial arts in the sensory and affective dimensions of the SF-MPQ, the VAS and PPI. Brazilian Jiu Jitsu athletes reported a higher prevalence of sensory and emotional dimensions of pain compared to Muay Thai and Boxing athletes due to the unique impact of Brazilian Jiu Jitsu techniques on pain experiences. Also, they have experienced more pain as measured by VAS and PPI than the Muay Thai and Boxing athletes. In contrast, the Boxing athletes had the lowest rating of pain among the three martial arts. It is unclear why Brazilian Jiu Jitsu athletes had greater pain than the other two martial arts; it is possible that the characteristics of their technique may be responsible for this result. More research in the future should confirm the results of this study. Finally, a relationship between age and pain perception was found in Brazilian Jiu Jitsu and Boxing athletes, but no other study has investigated it. Thus, more studies should confirm this result.

This is the first study that examines LBP, pain perceptions and disability due to pain between Brazilian Jiu Jitsu, Muay Thai and Boxing athletes. The present study used validated and reliable instruments and a large sample of martial arts athletes. Trainers and coaches of the martial arts should understand the characteristics of CLBP, the athlete’s pain beliefs and disability perception in each martial art to develop strategies to prevent CLBP. When examining martial arts athletes’ pain perceptions and disability, rehabilitation personnel should apply techniques that may reduce their pain so as to increase their sport performance. In particular, the trainers and coaches should modify athletes’ training characteristics so as to help them control their pain and improve their sport performance. Specifically, regarding the prevention of pain occurrence, developing a training plan that includes strength training, neuromuscular coordination, core stabilization exercise and cognitive behavioral techniques will help decrease the incidence of low back pain. Regarding the treatment of pain, due to the fact that the time spent on recovery may be short and repeated overloading is applied to the lumbar spine, LBP affects athletic performance. Recent clinical practical guidelines confirm the previous ones and demonstrate the need to use non-pharmacological and pharmacological treatments in the management of acute, subacute and chronic LBP. Key recommendations are placed on active treatments, including education, exercise, staying active, avoiding bed rest and self-management. Studies have reported that athletes with LBP may undergo a general physical therapy intervention that has long been used for pain control or that they may need to take time off from training to facilitate recovery [34,35]. Guidelines also encourage treatments targeting psychosocial and physical factors [36]. There is a possibility of our sample showing low pain due to their psychological strength and their high pain tolerance as athletes of martial arts. Further research should confirm the present results.

However, the present study has many limitations. Firstly, it is not feasible to generalize the results to other types of martial arts such as Karate and Judo. Another limitation is the cross-sectional design of this study, which carries with it certain limitations. The major limitation is the possibility of survival bias. Although disability was low in our sample, it is possible that the athletes with higher disability levels had abandoned training sessions and were not present at the time of the researcher’s visits. This could lead to underestimation of disability due to LBP. It is possible that athletes who were interviewed presented pain of low intensity that allowed the practice to continue. Also, the timing of athlete’s pain assessment may influence the results of the present study. If the questionnaires were completed at the end of the training session or during the competition period, the results may be different and the athletes may feel more LBP. Another limitation is related to pain. We did not investigate in depth the association of pain characteristics with the exact athletes’ movements such as frequency and duration of Jiu Jitsu, Boxing or Muay Thai movements that make pain worse. Considering the sample, we assessed only recreational athletes and not professional athletes. Also, we did not examine the impact of LBP on sport performance. Furthermore, the study failed to consider the potential impact of protective equipment, such as waist bandages, on the reported levels of pain and functional status.

More research should investigate the impact of the demographical athletes’ characteristics such as sex and years of experience in martial arts on LBP, perceptions and disability. Older athletes with more years of experience may be more able to control their pain, and thus have better pain tolerance and not lose any training or competition sessions. How the pain, pain perception and disability of martial arts practitioners is affected by the onset of training should be clarified in the future studies. If athletes had pain before starting training, they may differ in pain perceptions and disability with those whose pain starts after training. Finally, a significant theoretical and clinical question that should be answered is “could sport training reduce pain and improve health-related quality of life in martial arts?” We need more investigation into this research topic, and thus future studies are needed to cover these limitations and investigate martial arts, LBP and disability in relation to athletes’ sport performance.

## 5. Conclusions

Our findings suggest that Brazilian Jiu Jitsu, Muay Thai and Boxing athletes do not show pain and disability due to LBP. Also, these athletes reported that the pain does not last long and does not interfere with their daily activities. However, between the three martial arts athletes, Brazilian Jiu Jitsu athletes showed greater risk of experiencing pain, greater disability due to pain and greater pain beliefs than Muay Thai and Boxing athletes. Future studies are needed to confirm the present results and examine the factors that contribute to the appearance of LBP and if LBP affects their sport performance. Lastly, it is necessary to investigate differences in demographic characteristics between martial arts practitioners with LBP and if physical activity reduces pain and improves health-related quality of life in martial arts.

## Figures and Tables

**Table 1 healthcare-13-00447-t001:** Sample demographics (*n* = 90).

Characteristics		*n*	%
Gender	Man	46	51.1%
	Woman	44	48.9%
Education	Primary school	3	3.3%
	High school	17	18.9%
	University	49	54.4%
	Other	21	23.3%
Marital status	Unmarried	75	83.3%
	Married	15	16.7%
Kids	No	78	86.7%
	1	7	7.8%
	2	5	5.6%
Sport experience	<1 year	13	14.4%
	1–3	35	38.9%
	3–5	15	16.7%
	5–8	10	11.1%
	8–10	8	8.9%
	>10	9	10.0%
Training frequency	1–2/week	18	20.0%
	3–4/week	37	41.1%
	5/week	25	27.8%
	Daily	10	11.1%
Training duration	<1 h/day	14	15.6%
	2 h/day	64	71.1%
	>2 h/day	12	13.3%
Having low back pain	6–8 months	29	32.2%
	8–10 months	19	21.1%
	10–12 months	14	15.6%
	1–2 years	8	8.9%
	2–4 years	12	13.3%
	>4 years	8	8.9%
Missing training (month)	No	58	64.4%
	1	10	11.1%
	2	14	15.6%
	3	3	3.3%
	4	3	3.3%
	5	2	2.2%

**Table 2 healthcare-13-00447-t002:** Sample demographics by sport.

		Sport					
		Muay Thai		Brazilian Jiu Jitsu		Boxing	
		f	%	f	%	f	%
Gender	Man	15	50.0%	17	56.7%	14	46.7%
	Woman	15	50.0%	13	43.3%	16	53.3%
Education	Primary school	0	0.0%	1	3.3%	2	6.7%
	High school	5	16.7%	8	26.7%	4	13.3%
	University	18	60.0%	17	56.7%	14	46.7%
	Other	7	23.3%	4	13.3%	10	33.3%
Marital status	Unmarried	24	80.0%	24	80.0%	27	90.0%
	Married	6	20.0%	6	20.0%	3	10.0%
Sport experience	<1 year	3	10.0%	5	16.7%	5	16.7%
	1–3	11	36.7%	12	40.0%	12	40.0%
	3–5	3	10.0%	3	10.0%	9	30.0%
	5–8	5	16.7%	5	16.7%	0	0.0%
	8–10	4	13.3%	3	10.0%	1	3.3%
	>10	4	13.3%	2	6.7%	3	10.0%
Training frequency	1–2/week	5	16.7%	6	20.0%	7	23.3%
	3–4/week	12	40.0%	16	53.3%	9	30.0%
	5/week	9	30.0%	4	13.3%	12	40.0%
	Daily	4	13.3%	4	13.3%	2	6.7%
Training duration	<1 h/day	2	6.7%	2	6.7%	10	33.3%
	2 h/day	23	76.7%	26	86.7%	15	50.0%
	>2 h/day	5	16.7%	2	6.7%	5	16.7%
Having low back pain	<6 months	0	0.0%	0	0.0%	0	0.0%
	6–8 months	6	20.0%	10	33.3%	13	43.3%
	8–10 months	8	26.7%	7	23.3%	4	13.3%
	10–12 months	4	13.3%	4	13.3%	6	20.0%
	1–2 years	3	10.0%	3	10.0%	2	6.7%
	2–4 years	4	13.3%	4	13.3%	4	13.3%
	>4 years	5	16.7%	2	6.7%	1	3.3%
Missing training (month)	No	19	63.3%	14	46.7%	25	83.3%
	1	4	13.3%	4	13.3%	2	6.7%
	2	5	16.7%	6	20.0%	3	10.0%

**Table 3 healthcare-13-00447-t003:** Descriptive statistics of the Pain Perceptions Beliefs Inventory, the Quebec Back Pain Disability Scale and the Short-Form McGill Pain Questionnaire.

	Mean	Standard Deviation	Minimum	Maximum
Pain Perceptions Beliefs Inventory	−1.43	2.23	−6.30	4.45
Mystery	−0.18	0.83	−2.00	2.00
Self-blame	−0.34	1.10	−2.00	2.00
Duration	−0.82	0.98	−2.00	1.75
Stability/permanence	−0.36	0.76	−1.80	1.80
Quebec Back Pain Disability Scale (Total)	18.98	22.71	0	86
Bed/rest	1.93	2.27	0	9.00
Sit/stand	2.59	3.12	0	11.67
Ambulation	1.76	2.45	0	9.67
Handling heavy objects	3.22	4.28	0	18.00
Movement	1.06	1.48	0	6.50
Bending	2.10	3.18	0	11.25
Short-Form McGill Pain Questionnaire (Total)	12.34	8.91	2.00	38.00
Sensory	9.13	6.49	2.00	29.00
Emotional	3.21	2.90	0.00	10.00
VAS scale	1.65	1.82	1.00	9.20
Present Pain Intensity Index	2.10	1.08	1.00	5.00

**Table 4 healthcare-13-00447-t004:** Differences between Muay Thai, Brazilian Jiu Jitsu and Boxing athletes in the Pain Perceptions Beliefs Inventory, the Quebec Back Pain Disability Scale and the Short-Form McGill Pain Questionnaire.

	Sport						df, Kruscal- Wallis H	*P*
	Muay Thai		Brazilian Jiu Jitsu		Boxing			
	Mean	Standard Deviation	Mean	Standard Deviation	Mean	Standard Deviation		
Pain Perceptions Beliefs Inventory (Total)	−1.52	1.96	−0.54	2.32	−2.22	2.15	2, 9.32	0.009 **
Mystery	−0.28	0.77	0.18	0.98	−0.44	0.60	2, 7.40	0.025 *
Self-blame	−0.23	1.06	−0.32	1.00	−0.48	1.26	2, 1.24	0.537
Duration	−0.93	0.64	−0.38	1.18	−1.15	0.91	2, 9.34	0.009 **
Stability/permanence	−0.37	0.68	−0.08	0.79	−0.63	0.74	2, 9.52	0.009 **
Quebec Back Pain Disability Scale (Total)	13.80	13.42	35.57	29.73	7.57	8.42	2, 25.33	<0.001 ***
Bed/rest	1.62	1.84	3.33	2.77	0.82	1.15	2, 20.06	<0.001 ***
Sit/stand	2.43	3.08	4.17	3.73	1.17	1.33	2, 11.11	0.004 **
Ambulation	1.24	1.80	3.22	2.89	0.82	1.85	2, 17.75	<0.001 ***
Handling heavy objects	2.26	3.31	5.81	5.22	1.60	2.27	2, 15.04	<0.001 ***
Movement	0.75	0.98	1.76	1.80	0.66	1.33	2, 12.43	0.002 **
Bending	1.41	2.41	4.09	4.06	0.79	1.55	2, 16.19	<0.001 ***
Short-Form McGill Pain Questionnaire (Total)	14.63	13.42	22.37	25.78	12.47	10.65	2, 1.84	0.399
Sensory dimension	2.90	3.11	4.80	2.93	1.93	1.78	2, 15.73	<0.001 ***
Emotional dimension	11.20	7.70	17.87	9.87	7.97	5.86	2, 19.34	<0.001 ***
Visual Analogue Scale	2.13	1.14	2.83	0.91	1.33	0.55	2, 34.95	<0.001 ***
Present Pain Intensity Index	0.83	0.51	1.31	0.73	0.60	0.47	2, 18.60	<0.001 ***

* *p* < 0.05, ** *p* < 0.01, *** *p* < 0.001.

## Data Availability

The dataset is available upon request from the corresponding author.

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
