# Peer review of "Examination of Non-Specific Low Back Pain, Pain Perceptions and Disability Between Brazilian Jiu Jitsu, Muay Thai and Boxing Athletes"

_healthcare, 2025, doi:10.3390/healthcare13050447_

Round 1
Reviewer 1 Report (Previous Reviewer 2)
Comments and Suggestions for Authors
It is a bit hard to follow how the authors have responded to my previous comments, so I might have missed some of them.
One thing I reacted on was the inclusion criteria d, diagnosed with chronic ... pain by a physician using MRI-scan. The authors responded that they should change this since pain is not visible on MRI-scan, but it still is there. Do they mean that MRI-scan is used as exclusion criteria, where the physician using MRI-scan exclude different conditions visible on MRI-scan? If so, move this to exclusion criteria. MRI-scan has nothing to do with whether the patient has pain or not - you just ask the patient and you get the answer. And the chronicity is connected to for how long the pain have persisted not if the MRI-scan shows anything.
About the title. I still think the last words: "with chronic low back pain" ought to be excluded from the title since a pain level of 16 where maximum is 100 can not be considered as "chronic pain". When treating pain, if a patient arrives and say that the current pain level is 16/100, the patient is dismissed and asked to return when their pain is at least 40/100. If the authors really want to examine chronic pain patients, they need to exclude those with lower pain levels than 30/100. Otherwise it is like studying "perception and disability among patients with hypertension" and include subjects without medication and with normal blood pressure.
There are aspects that could be explored in this study and hence make it interesting. For example, how was pain perception related to age? Where there subgroups of subjects that had pain before they started training, and how was the pain affected by this? Were there differences in pain between athletes that had newly started training and those that had been performing for several years (decades?) In current form, the manuscript might be OK to publish, but it adds very little to the current knowledge. In a revised form it could be very interesting, especially if it assess the question: Could physical activity in form of martial arts reduce pain and improve health-related quality of life?
However, it is interesting
Author Response
Dear Reviewer
We would like to thank again the Reviewer 1 for his/her time for making a constructive feedback to improve our manuscript. Your suggestions and corrections have been done accordingly (green highlight inside the 2nd time resubmitted manuscript, yellow highlight from the previous version of resubmitted manuscript) and we hope that the manuscript is appropriate now for publication to Healthcare.
Best regards

Reviewer 2 Report (New Reviewer)
Comments and Suggestions for Authors
Reviewer Comments:
1. Introduction:
- Indicate the duration of pain when referring to chronic pain.
- Definition of chronic low back pain and differences with other types of pain, as well as differences with acute and chronic pain.
- Further contextualization of chronic low back pain and the influence of psychosocial factors in this type of patient would be appropriate.
- The authors state that there is a high level of research in other sports. Which of these could be included and which evaluated. In this way the research problem could be better clarified.
2. Materials and Methods
- how did participants gain access to the study?
- Who made the diagnosis of chronic low back pain? Was it the study investigating physician or was it a physician outside the study?
- The authors established as an inclusion criterion: diagnosis by magnetic resonance imaging. According to scientific evidence, it has been shown that pain does not necessarily have to be related to structural damage diagnosed by imaging tests. How can the authors explain this? Why did they establish the diagnosis by magnetic resonance imaging? What did they want to see in these imaging tests?
- Patients included in the study could also have pain radiating down the legs? Is it different from neuropathic pain?
-at what point were the athletes assessed?
3. Discussion
-The influence of the timing of athlete assessment could be discussed. At the moment of competition or training they are in, as this could influence the results.
- In addition to addressing some more practical clinical applications in relation to the study objective
- Some more references should be included, especially related to chronic and current low back pain. Latest clinical practice guidelines for example. In addition to other studies to contribute to the introduction and discussion to contextualize and discuss the main objectives and results of the study.
Author Response
Dear Reviewer
We would like to thank the Reviewer for his/her time for making a constructive feedback to improve our manuscript. Your suggestions and corrections have been done accordingly (green highlight inside the 2nd time resubmitted manuscript, yellow highlight from the previous version of resubmitted manuscript) and we hope that the manuscript is appropriate for publication to Healthcare.
Best regards

Round 2
Reviewer 1 Report (Previous Reviewer 2)
Comments and Suggestions for Authors
Thank you for your revision. I will take it step by step.
In the first sentence in the introduction the authors state: "LBP... accounts for approximately 90% of LBP" - which is taken from an article by Wirth et al, who have taken it from an article by Koes et al from 2006 that have based it on previous literature, and where pain is defined as chronic after 6 weeks duration. It was also based on assessment of patients in primary healthcare with pain, and since more specific pain often is managed in secondary care the figures are very uncertain. In current literature non-specific LBP is closer to 50-60%. As the figure is not important for the study, just omit the part of the sentence with 90%.
About categorization, the current definition of chronic is 12 weeks or more as the authors state later in the paragraph. I don't understand what the sentence on line 34 adds.
Line 45: This is true for both acute and chronic LBP.
As I said in the previous review: The manuscript would be (and now is) OK in current version after revision, but it does not add much to the knowledge. If you not only mentioned my suggestions in the limitations and "further research" section and instead expanded the study to include those kinds of results in the study you would have improved the rating immensely. Now I can't but rate it as average.
Author Response
Dear Reviewer
We would like to thank you once more time for your constructive feedback in order to improve our manuscript. Your corrections have been done (grey highlight inside the manuscript). We hope that the manuscript is now appropriate for publication to the Healthcare (Special Edition: Common Sports Injuries and Rehabilitation).
Best regards
The corresponding author

Reviewer 2 Report (New Reviewer)
Comments and Suggestions for Authors
Most of the reviewer's comments have been addressed.
Author Response
Dear Reviewer
Thank you!
Best regards
The corresponding author
This manuscript is a resubmission of an earlier submission. The following is a list of the peer review reports and author responses from that submission.
Round 1
Reviewer 1 Report
Comments and Suggestions for Authors
Dear Authors,
Thank you for submitting your manuscript for review. The study addresses a relevant and underexplored topic, comparing chronic low back pain and related perceptions among athletes practicing different martial arts disciplines. However, after a detailed review, I have identified several areas where the manuscript requires substantial improvements in terms of clarity, rigor, and interpretation. Below, I provide specific comments and suggestions for each section of the manuscript:
Abstract:
The opening sentence is very brief and does not adequately contextualize the study. I recommend either removing it or improving the background to provide a stronger introduction to the research problem.
Statements such as “rated their pain as mild” and “Muay Thai athletes experienced a greater intensity of pain” may confuse readers. I suggest specifying which scale or metric was used to assess these pain levels and clarifying whether the values refer to absolute measurements or relative differences between groups.
It would be helpful to include a brief mention of how the findings might be applied in clinical practice or in designing specific training programs to reduce the risk of chronic low back pain in these athletes.
Introduction:
The introduction lacks a logical transition that connects the general issue of chronic low back pain to martial arts athletes specifically. I recommend restructuring this section to build a clearer bridge between the problem and the target population, thereby better justifying the relevance of the study.
Materials and Methods:
The study design is not sufficiently described in this section, which is crucial for assessing methodological rigor.
The participant selection process also lacks detail. I recommend including additional information to clarify the inclusion and exclusion criteria, as well as how participants were recruited.
A flowchart illustrating the selection process and study development would greatly improve the clarity of this section.
Results:
The statement that athletes rated their pain as “mild” (M=12.34, SD=8.91) appears contradictory when it is later mentioned that Muay Thai athletes experienced greater pain intensity compared to other groups. This needs clarification: is the pain mild in absolute terms, or are the results relative between the groups?
The tables accompanying the results lack clear interpretation in the text. I recommend explicitly discussing the data presented in the tables, highlighting the most relevant findings to help readers better understand their implications.
Discussion:
The discussion section largely reiterates the results without providing in-depth interpretation or meaningful connections to the existing literature. Comparisons with other studies are scarce and seem disconnected from the objective of the study.
The current version of the manuscript has limitations that affect its clarity and methodological rigor, which should be improved. Addressing these issues would require extensive revisions. I encourage the authors to revisit the study's design, analysis, and interpretation to better align it with the standards of scientific research.
Reviewer 2 Report
Comments and Suggestions for Authors
The introduction made me interested in the subject, thinking this is a new kind of prevalence studies connected to the participants activities. I could follow the authors line of thoughts all through introduction and methods, but when I came to results I got confused. There are only different tables and not a clear description of the results. Then again, in the discussion some results where presented mixed with ideas presented in previous articles by other authors. It was thus hard to understand what the results of the study were.
In several paragraphs the authors write that the "participants experienced their pain as mild". It made me wonder if this really was participants with low back pain. The inclusion criteria was said to be "diagnosed the chronic low back pain by a physician with a MRI scan" which is a strange statement since pain is not visible on MRI scan - pain is something that is perceived by a person (patient, participant). When I had read the discussion and the statement that the participants description of mild pain was repeated I tried to find any quantitative measure of the pain level. In table 1 (both the first Table 1 and the second (?) table 1) nothing is said about the pain level. In table 2 they describe pain level in visual analogue scale (VAS) to be 16.49 for the whole group (which is very low and usually not considered to be described as pain in any clinical aspect. VAS is measured having a range of 0-100). In table 3 they divide the three groups and give measures of VAS 2.13, 2.83 and 1.33 which is extreme low! These values of VAS don't only state that this is not a group of patients with low back pain that they have studied, they also make clear that there is an inconsistency between measures for the whole group and for the 3 separate subgroups.
If the authors totally rewrite the manuscript, changing the title to "Examination of the pain, pain perceptions and disability between Brazilian Jiu Jitsu, Muay Thai and Boxin athletes" and focus on how the different athletes perceive pain and think about pain the manuscript might be possible to publish as a description of how athletes think. In the present form the manuscript gives false answers.
